

**PeerJ Hubs**

Published on behalf of

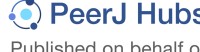


# Feeding preferences and the effect of temperature on feeding rates of the graceful kelp crab, *Pugettia gracilis*

Katrina H. Johnson[1,2,3], Katie A. Dobkowski[1,2,4], Sasha K. Seroy[2,5], Shelby Fox[2,5] and Natalie Meenan[1]

[1] Bates College, Lewiston, ME, United States of America
[2] Friday Harbor Laboratories, University of Washington, Friday Harbor, WA, United States of America
[3] Scripps Institution of Oceanography, La Jolla, CA, United States of America
[4] Woodbury University, Burbank, CA, United States of America
[5] University of Washington, School of Oceanography, Seattle, WA, United States of America

Corresponding author
Katrina H. Johnson,
khjohnson17@gmail.com

## ABSTRACT

Graceful kelp crabs (*Pugettia gracilis*) are abundant consumers in shallow subtidal ecosystems of the Salish Sea. These dynamic habitats are currently experiencing multiple changes including invasion by non-native seaweeds and ocean warming. However, little is known about *P. gracilis'* foraging ecology, therefore we investigated their feeding preferences between native and invasive food sources, as well as feeding rates at elevated temperatures to better assess their role in changing coastal food webs. To quantify crab feeding preferences, we collected *P. gracilis* from San Juan Island, WA and conducted no-choice and choice experiments with two food sources: the native kelp, *Nereocystis luetkeana,* and the invasive seaweed, *Sargassum muticum*. In no-choice experiments, *P. gracilis* ate equal amounts of *N. luetkeana* and *S. muticum*. However, in choice experiments, *P. gracilis* preferred *N. luetkeana* over *S. muticum*. To test effects of temperature on these feeding rates, we exposed *P. gracilis* to ambient (11.5 ± 1.3 °C) or elevated (19.5 ± 1.8 °C) temperature treatments and measured consumption of the preferred food type, *N. luetkeana*. Crabs exposed to elevated temperatures ate significantly more than those in the ambient treatment. Our study demonstrates the diet flexibility of *P. gracilis*, suggesting they may be able to exploit increasing populations of invasive *S. muticum* in the Salish Sea. Warming ocean temperatures may also prompt *P. gracilis* to increase feeding, exacerbating harmful impacts on *N. luetkeana,* which is already vulnerable to warming and invasive competitors.

## INTRODUCTION

Crabs play significant ecological roles in coastal food webs. In a variety of coastal ecosystems, crabs can play an important trophic role as grazers capable of modulating populations of foundational primary producers. In kelp habitats, crab herbivory mediates kelp growth and survival (*Dobkowski, 2017*), kelp forest density (*Jofré Madariaga, Ortiz & Thiel, 2013*), and kelp community structure and productivity (*Gaines & Lubchenco, 1982*; *Dobkowski*

*et al., 2017*; *Dobkowski, 2017*). In salt marshes, shore crabs also serve as key consumers and regulators of salt marsh vegetation and biomass (*Beheshti et al., 2021*).

Kelp crabs (genus *Pugettia*), often feed on brown macroalgae and occur in the same range in latitude and longitude as bull kelp (*Nereocystis luetkeana*), from Alaska to California (*Morris, Abbott & Haderlie, 1980*). They are common residents of *N. luetkeana* beds in the Salish Sea, which range from rocky shallow subtidal areas to 30 m depth (*Kruckeberg, 1991*; *Duggins, Simenstad & Estes, 1989*). The graceful kelp crab, *Pugettia gracilis,* and the Northern kelp crab, *Pugettia producta,* are significant consumers of *N. luetkeana* (*Dobkowski, 2017*; *Dobkowski et al., 2017*), along with sea urchins (genus *Strongylocentrotus* and *Mesocentrotus;* (*Paine & Vadas, 1969*) and snails (*Lacuna vincta*; (*Chenelot & Konar, 2007*). Although *P. gracilis* is known to be found in kelp beds in the Salish Sea (*Dobkowski, 2017*), little has been quantified about their actual feeding preferences. Understanding these effects on *N. luetkeana* is important as it is a foundation species that provides habitat and serves as a food source for many marine fishes, mammals, and invertebrate species in the Pacific Northwest of North America (*Steneck et al., 2002*; *Carney et al., 2005*; *Daly & Konar, 2010*). Although an important part of the ecosystem, grazers and herbivores are not the only pressure being placed on native kelps.

*Sargassum muticum* is an invasive species and a potential competitor to native kelps, including *N. leutkeana* (*Britton-Simmons, 2004*; *Gaydos et al., 2008*). On the west coast of North America, *S. muticum* is now found from Alaska to Mexico, overlapping in range with the distributions of *N. leutkeana* and *P. gracilis*. In the early 20th century, *S. muticum* was introduced to Washington State and the Salish Sea from the Western Pacific (*Britton-Simmons, 2004*; *Seebach, Colnar & Landis, 2010*). Along the west coast, invasive *S. muticum* has demonstrable negative impacts on native seaweeds through direct competition and inhibition of recruitment (*Ambrose & Nelson, 1982*; *DeWreede, 1983*; *Stæhr et al., 2000*). In the Salish Sea specifically, *S. muticum* populations have been shown to reduce light and nutrient concentration availability for native kelp species, reducing the native canopy by 75% and the understory by 50% (*Britton-Simmons, 2004*). Increased populations of *S. muticum* have also been documented to influence grazers within subtidal ecosystems. Experiments demonstrate that *S. muticum* serves as an additional food supply, shelter, and spawning habitat for many crab and snail species (*Seebach, Colnar & Landis, 2010*; *Britton-Simmons et al., 2011*), while other species such as the green sea urchin, *Strongylocentrotus droebachiensis*, are deterred by *S. muticum* (*Marks, Reed & Holbrook, 2020*).

Coastal food webs of the Salish Sea are experiencing both short-term and long-term warming due to climate change. The annual mean water temperature in the Salish Sea is 10 °C, fluctuating between 6.5 °C to 13 °C seasonally (*Khangaonkar et al., 2019*). Climate models predict that by 2100 Salish Sea surface temperatures will increase by approximately 1.57 °C and estuarine and intertidal ecosystems, the ecosystems in which kelp crabs are found, warming more intensely by 3.23 °C (*Khangaonkar et al., 2019*; *Berry et al., 2021*). Short-term, intense periods of warming due to atmospheric heatwaves are also becoming more common and have created large-scale shifts in the structure of kelp forests (*Berry et al., 2021*; *Khangaonkar et al., 2021*; *McPherson et al., 2021*; *Raymond et al., 2022*). Atmospheric
heatwaves are short time periods (2+ days) of elevated air temperatures, based on historical averages for a given area, that influence shallow nearshore environments (*Raymond et al., 2022*). Atmospheric heatwaves are most influential and relevant in intertidal and nearshore ecosystems (*Raymond et al., 2022*). In 2014–2016, Salish Sea estuarine temperatures were warmer by an average of 2.3 °C (*Khangaonkar et al., 2021*). Nearshore intertidal sea surfaces were warmer by a maximum of 6.2 °C (*Gentemann, Fewings & García-Reyes, 2017*). Intertidal organisms often experience a wide range of temperatures with short term high temperature exposures within the range of elevated temperatures tested within this experiment. Temperature changes are often accompanied by other abiotic changes, like changes in salinity or acidification that may also influence biological responses to environmental stress.

Food web interactions and the organisms within them have been influenced by warming temperatures in a variety of ways. Many herbivores, including crabs, have shown increased feeding at high temperatures due to an increase in metabolic requirements (*Hill, 1980*; *McPherson et al., 2021*). For example, in laboratory studies the mud crab, *Scylla serrata*, increased feeding at elevated temperatures of 20 °C and 25 °C (*Hill, 1980*). The same trend of consuming more at higher temperatures has also been shown in king crabs, *Paralithodes camtschaticus*, (*Siikavuopio & James, 2015*) and shore crabs, *Carcinus maenas* (*Elner, 1980*). In simulations based on the 2014–2016 northeast Pacific heatwave, there was an increase in biological activity due to ocean temperatures (*Khangaonkar et al., 2021*). Thus, understanding the effects of herbivores like *P. gracilis*, warming, and interactions on both *N. luetkeana* and *S. muticum* is especially critical.

As ocean temperatures increase on both long-term and episodic timescales in the Salish Sea, it is of particular importance to understand how ecologically significant organisms and the food webs they comprise may respond. The sensitivity of adult *N. luetkeana* to ocean warming has been well documented and linked to recent declines in many Pacific Northwest and Salish Sea populations (*Schiel, Steinbeck & Foster, 2004*; *Supratya, Coleman & Martone, 2020*; *Berry et al., 2021*). In laboratory studies, the growth and development of healthy *N. luetkeana* spores had reduced success when incubated in temperatures of 17 °C or above (*Schiltroth, Bisgrove & Heath, 2018*). It remains unclear how ocean warming affects the ability for *N. luetkeana* to respond to other environmental factors, such as invasive species like *S. muticum.* Increased *S. muticum* populations have been correlated with rising seawater temperatures and increased nutrient enrichment (*Norton, 1977*; *Wang et al., 2019*). Increasing populations of *S. muticum* may influence the distribution and abundance of *N. luetkeana* and consequently associated food web interactions. If this continues, *N. luetkeana* populations could experience this additional stressor in parallel to their already documented vulnerabilities to warming (*Schiel, Steinbeck & Foster, 2004*; *Supratya, Coleman & Martone, 2020*).

Our first objective in this study was to understand the relationship between *P. gracilis* and both native and invasive food sources. We asked: (1) Can *P. gracilis* consume both *N. luetkeana* and *S. muticum*? And if so, (2) does *P. gracilis* have a preference between these two food sources? We hypothesized that *P. gracilis* would be able to consume both seaweeds but would prefer the native *N. luetkeana* over the invasive *S. muticum* because

similar feeding preferences have been demonstrated by the closely related Northern kelp crab (*Pugettia producta*) (*Dobkowski et al., 2017*). We conducted choice and no-choice feeding trials to determine what seaweeds *P. gracilis* can eat as well as quantify which food sources they prefer to eat when given two options. By studying these feeding preferences, we can better understand the future impacts of invasive species on *P. gracilis* and the ecological pressures that potentially affect *N. luetkeana*.

Our second objective was to assess how *P. gracilis* feeding is affected by elevated temperatures characteristic of short-term warming subtidal ocean conditions in the Salish Sea. We hypothesized that *P. gracilis* would increase feeding rates at higher temperatures similar to observed trends in other invertebrate species (*Elner, 1980*; *Carr & Bruno, 2013*; *Siikavuopio & James, 2015*). To do this, we conducted feeding trials (using the preferred food option, *N. luetkeana*) at two temperatures, ambient (11.5 ± 1.3 °C) and elevated (19.5 ± 1.8 °C), to determine the differences among feeding rates between temperatures.

## METHODS

### No-choice and choice experiments

We collected 12 *P. gracilis* (mean and SD: 3.5 ± 1.8 g) from 0 to 2 m depth from four sites in June and July of 2020 (Reuben Tart County Park, Deadman's Bay, Friday Harbor Labs, and Eagle Cove; Supplemental 1, Table S1). Crabs were collected randomly without bias towards sex or size. We did not use ovigerous females and all crabs were adult crabs. Once collected, we housed the crabs in flow-through seawater tanks at ambient water temperature (11−12 °C) at Friday Harbor Labs (FHL). Crabs resided individually in plastic tanks (28 cm × 15 cm × 11 cm) with mesh lids and consumed a mixed diet of local seaweeds prior to, and in between, experiments. We used a block experiment design, treating the crabs as the blocking factor, to examine the feeding preferences of *P. gracilis*. Two of the twelve crabs molted during the experiment and therefore we have only utilized the data from before the molting event.

We conducted two sets of experiments: (1) no-choice feeding experiments, where crabs were offered only one food type (*N. luetkeana* or *S. muticum*), and (2) choice feeding experiments, where crabs were offered both food types simultaneously. No-choice experiments enabled us to quantify food consumption of each food type individually while the choice experiments enabled us to assess food preference between the two food types. Each crab participated in three independent feeding trials in a randomized order: *N. luetkeana* only, *S. muticum* only, and choice of *N. luetkeana* and *S. muticum*, leading to a total of 36 trials (Supplemental 2).

To prepare food for these experiments, we collected *N. luetkeana* and *S. muticum* from detached, floating "drift" sources near FHL. All food sources were harvested from the same drift patch and we used only non-reproductive kelp blades to standardize for freshness of the food source and environmental factors such as drift time. The wet mass of food items was determined prior to every trial using an Ohaus Navigator XT scale. We offered similar sized pieces of *N. luetkeana* (5.52 ± 0.33 g) and *S. muticum* (5.55 ± 0.43 g) to avoid biasing the crabs toward the visibly larger food item.
Each trial lasted three days (Supplemental 2). Crabs spent the same amount of time in the flow-through tanks prior to the experiment. Crabs were starved for 24 h prior to each independent feeding period. The starvation period was followed by a 48 h feeding period which took place in the same flow-through seawater tanks holding the crab. Pilot experiments showed 48 h to be sufficient time for crabs to eat a quantifiable amount of the food sources offered. We used a short feeding time to minimize effects of nutrient enhancement due to crab excretion on seaweed mass. We measured blotted wet mass before and after each experiment to calculate how much each crab consumed. Thirty-six controls used the same experimental design, which included no crabs, to account for natural mass loss or gain of seaweed not due to crab consumption over a 48 h period. Each control was run and subtracted from the corresponding trial.

We conducted a Shapiro Test for normality (choice: $p = 0.4126$; no-choice: $p = 0.9729$) and therefore proceeded with a Bartlett Test for normally distributed data to test for equal variances across test groups (choice: $p = 0.463$; no-choice: $p = 0.5846$). All conditions were met to proceed with a $t$-test for differences in means. To analyze these data, we used an unpaired $t$-test for the no-choice experiments and a paired $t$-test for the choice experiments to determine if there were significant differences in food consumption between the food types. We calculated algal mass consumed (g) per crab mass (g) for comparability purposes. We did not assess effects of sex or size because of the small sample size and limited power to include these factors as covariates. All analyses were conducted in R version 1.2.1335 (R Foundation for Statistical Computing Platform 2020).

## Temperature experiments

We collected 18 additional *P. gracilis* (mean and SD: 3.3 ± 1.2 g) from Friday Harbor Labs in November of 2020 and trials were run from November 2020 to February 2021 (Table S2). We did not hold or reuse crabs between feeding preference experiments and temperature experiments and ambient water temperatures where the laboratory seawater pump is located does not change appreciably throughout the year. We maintained crabs in the lab using the same set up described for the experiments above, and seaweeds were collected using the same procedure as in the feeding experiments.

Each of the crabs were randomly assigned to a temperature treatment: ambient (twelve *P. gracilis* mean and SD: 3.2 ± 1.1 g; temperature mean and SD: 11.5 ± 1.3 °C) or elevated (six *P. gracilis* mean and SD: 3.6 ± 1.3 g; temperature mean and SD: 19.5 ± 1.8 °C). Temperature in tanks was regulated using aquarium heaters. Temperature was monitored using a DS18B20 temperature probe which logged temperature every 10 min to a ESP8266 microcontroller running MicroPython. Crabs were placed in their own individual tanks with heaters. The unbalanced sample size was due to some elevated treatment crabs experiencing a significantly high temperature shock because of unstable heaters; thus those crabs were removed from the experiment. For elevated temperature treatments, the water began at ambient temperature and was slowly raised to reach the target temperature over the course of 24 h, a temperature increase that the study organisms could be exposed to in their natural environments on the same time scale. Crabs were acclimated to the elevated temperature treatment for another 48 h while feeding normally prior to trials. As in the

no-choice and choice experiments described above, crabs were then starved for 24 h before a 48 h feeding period (Supplemental 3). All crabs were given *N. luetkeana* only during feeding rate trials to control for food type. Eighteen controls (twelve in ambient water conditions, six in elevated water conditions) followed the same experimental design, which included no crabs, to account for natural mass loss or gain of seaweed not due to crab consumption. Each control was run and subtracted from the corresponding trial.

To compare the amount of *N. luetkeana* consumed between the two temperature treatments, we utilized a Welch's *t*-test to accommodate our unequal sample sizes. For analysis, we calculated algal mass consumed (g) per crab mass (g). This analysis was conducted in R version 4.2.2 (*R Core Team, 2022*).

## RESULTS

### No-choice and choice experiments

*P. gracilis* successfully consumed both *N. luetkeana* (mean and SE: $0.26 \pm 0.04$ g) and *S. muticum* ($0.22 \pm 0.03$ g) during the 48 h feeding period. Crabs displayed no significant difference in their consumption of *N. luetkeana versus S. muticum* when given no choice between the two food types ($t = 0.075$, $df = 10$ $p$-value = 0.464; Fig. 1).

Conversely, *P. gracilis* consumed significantly more *N. luetkeana* ($0.23 \pm 0.03$ g) than *S. muticum* ($0.15 \pm 0.04$ g) when given a choice between the two food types ($t = 3.038$, $df = 11$ $p$-value =0.013; Fig. 2).

### Temperature experiments

Crabs in the ambient temperature treatment consumed $0.14 \pm 0.05$ g of *N. luetkeana,* while crabs in the elevated temperature treatment consumed $0.58 \pm 0.10$ g over 48 h. Crabs exposed to elevated temperature ate significantly more *N. luetkeana* than those exposed to ambient temperature (Welch's t-test: t = $-2.564$, $df = 5.95$, $p$-value =0.04; Fig. 3).

## DISCUSSION

Our results show that *N. luetkeana* is the preferred food type of *P. gracilis* but also suggest that the crabs can modify their diet to exploit the invasive food source, *S. muticum*. This suggests *P. gracilis* has a more generalist diet beyond just the kelp for which the crabs are named. Generalist feeding strategies are common among crabs, including *Acanthonyx scutiformis,* a coastal crab and seaweed generalist, and *Uca annulipes,* another coastal crab and omnivore (*Milner et al., 2009*; *Vasconcelos et al., 2009*). Our finding that *P. gracilis* can eat equally as much *S. muticum* as *N. luetkeana* when given no choice is significant for their continued success in response to changing food availability which may be characteristic of future conditions. We did not assess effects of sex or size due to the small sample size but recognize that this is a variable to be considered in future studies. Further research is also needed to determine if both food types confer the same nutritional value. As a generalist consumer, *P. gracilis* is well-positioned to take advantage of an increasingly available invasive food source.

Our study is realistic of field food conditions because co-occurring *N. luetkeana* and *S. muticum* beds in the Salish Sea have been observed, emulating our choice experiments.
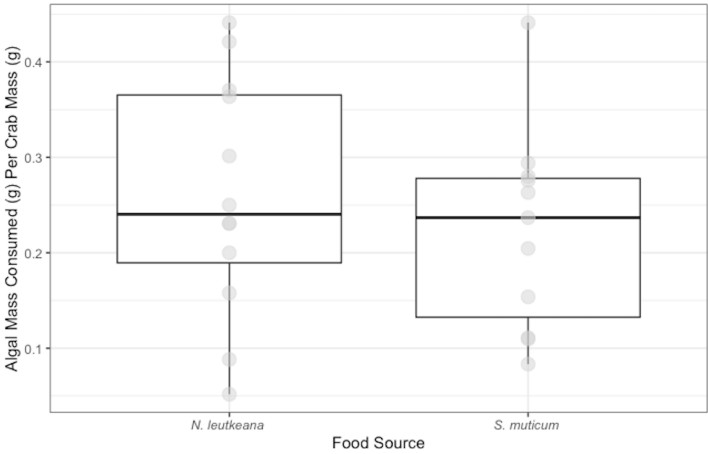

**Figure 1 No-Choice Experiment Feeding Levels of *P. gracilis*.** Medians and interquartile ranges of mass of seaweeds consumed (adjusted for the controls) by *P. gracilis* ($n = 12$) in no-choice experiments. Crabs consumed both types of food equally when they were not given options (unpaired t-test: $t = 0.075$, $df = 11$ p-value $= 0.464$).

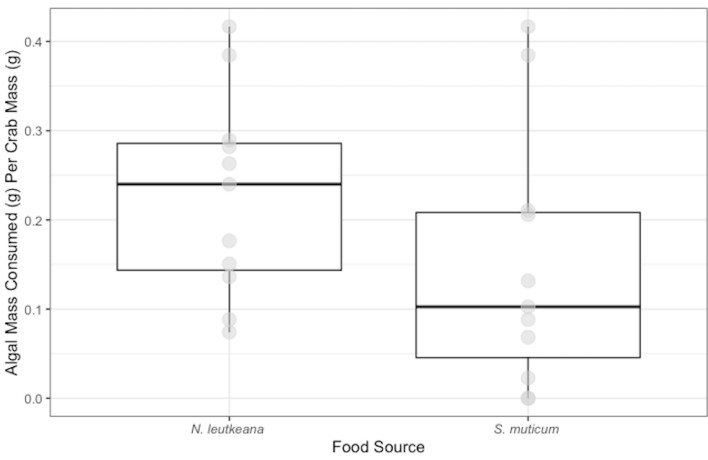

**Figure 2 Choice Experiment Feeding Levels of *P. gracilis*.** Medians and interquartile ranges of mass of seaweeds consumed (adjusted for the controls) by *P. gracilis* ($n = 12$) in choice experiments. *P. gracilis* consumed significantly more *N. luetkeana* than *S. muticum* when given the choice between the two food types (paired t-test: $t = 3.038$, $df = 11$ p-value $= 0.013$).

As *S. muticum* populations increase, food conditions recreated in our choice experiments may become more common, creating more opportunities for crabs to choose their food type. We have determined that *N. luetkeana* is the preferred food source of *P. gracilis* so in scenarios where *N. luetkeana* and *S. muticum* beds are co-occurring, even though *P. gracilis* can consume both, they may choose to consume *N. luetkeana*. Despite the increasing *S. muticum* populations and changing ocean temperatures, this feeding alone puts pressure on *N. luetkeana*.

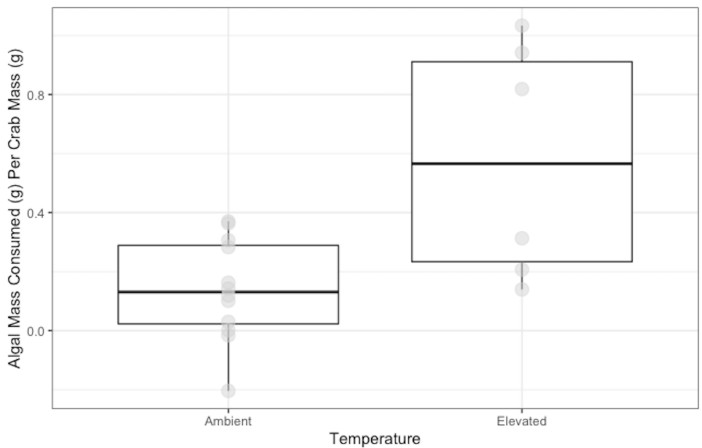

**Figure 3  Temperature Experiment Feeding Levels of *P. gracilis*.** Medians and interquartile ranges of mass of *N. luetkeana* consumed by *P. gracilis* ($n = 18$) in temperature treatments. The ambient temperature treatment had a mean of $11.5 \pm 1.3$ °C and the elevated temperature treatment had a mean of $19.5 \pm 1.8$ °C. Crabs in elevated temperature treatments consumed significantly more than those in ambient temperatures. (Welch's *t*-test, t $= -2.564$, *df* $= 5.95$, *p*-value $= 0.04$).

Our results suggest that at higher temperatures, *P. gracilis* consumes greater amounts of *N. luetkeana* than at ambient water temperatures. This response is consistent with increased consumption rates measured in other coastal invertebrates in response to increased temperatures. Green sea urchins, *Strongylocentrotus droebachiensis*, (*Carr & Bruno, 2013*), king crabs, *Paralithodes camtschaticus*, (*Siikavuopio & James, 2015*) and shore crabs, *Carcinus maenas* (*Elner, 1980*) have also exhibited increased consumption and metabolic rates at higher temperatures (*Carr & Bruno, 2013*). As ocean temperatures rise, our work demonstrates that *P. gracilis,* may increase the amount of food consumed, placing pressure on their preferred food source, *N. luetkeana*. Further research is needed to understand if *P. gracilis* increases consumption of their non-preferred food type, *S. muticum,* at elevated temperatures as well, and if these trends are consistent across closely related crabs. Though we only examined temperature as an abiotic effect on *P. gracilis* feeding, future studies should examine the effects of multiple co-occurring abiotic stressors.

*N. luetkeana* and other native kelp populations have demonstrated negative responses to elevated temperatures, whereas invasive *S. muticum* populations have demonstrated positive responses to similar elevated temperatures (*Ambrose & Nelson, 1982*; *DeWreede, 1983*; *Stæhr et al., 2000*; *Britton-Simmons, 2004*). In one study, high temperatures decreased growth and performance of native seaweeds, such as *Fucus serratus* and *Chondrus crispus,* but enhanced growth and performance of *S. muticum* (*Atkinson et al., 2020*). In the San Juan Islands, Washington, *S. muticum* survived in temperatures up to 28 °C, whereas *N. luetkeana* has decreased performance and fitness starting at 18 °C (*Lüning & Freshwater, 1988*). The future quantity and quality of *N. luetkeana* beds have been shown to, and is predicted to continue to, decrease with increasing ocean temperatures (*Simonson, Scheibling & Metaxas, 2015*; *Schiel, Steinbeck & Foster, 2004*; *Supratya, Coleman & Martone, 2020*). If

*P. gracilis* consumes more *N. leutkeana* at higher temperature, and *S. muticum* populations proliferate in warmer waters, future *N. luetkeana* survival may be threatened by these combined impacts.

Elevated temperatures used in our experiments have relevance to short term, episodic warming events that have become increasing more prevalent in intertidal ecosystems. Our elevated temperature conditions are aligned with the already established intertidal and shallow subtidal ecosystem warming predictions and mirror acute and episodic temperature stress due to the increasing prevalence of warming events like atmospheric heatwaves (*Raymond et al., 2022*). The short-term elevated temperature conditions in our experiment have the most near-term field relevance in the context of these episodic and extreme temperature stresses. With the short duration of temperature stress, six days, the temperature experiments closest resemble the time scale of recent atmospheric heatwaves and may predict the effects of a short temperature anomaly on food web interactions during these periods.

Many studies on ecological effects of warming and invasive species have been conducted independently, but recent evidence has shown that ocean warming and invasive species can act together to alter marine communities (*Stachowicz et al., 2002*; *Sorte, Williams & Carlton, 2010*; *Strayer, 2010*; *Miranda et al., 2019*; *Atkinson et al., 2020*). Our results show that in warmer waters, *S. muticum* may be increasingly able to outcompete *N. luetkeana*, supported by its increased productivity and growth rates in warmer waters (*Norton, 1977*; *Wang et al., 2019*). Therefore, both ocean warming and competition from invasive seaweeds each place separate stress on native seaweeds such as *N. luetkeana*. These stresses could be compounded by *P. gracilis'* preference for the native seaweed, *N. luetkeana*, over the invasive seaweed, *S. muticum* and increased consumption of *N. luetkeana* at elevated temperature. If this trend occurs in the field, *N. luetkeana* populations may experience compounding stressors in warming conditions from increased herbivory in addition to higher levels of competition from invasive seaweeds.

Graceful kelp crabs will likely be able to modify their diets as nearshore algal communities and food availability change in response to warming temperatures and invasive species. Our results show that *P. gracilis* eats more at higher temperatures but generalist feeding strategies make them well-positioned to manage changing ecosystems. Though we found that *S. muticum* was not the preferred food source of *P. gracilis*, it is a competitor of *N. luetkeana* which proliferates at high temperatures, suggesting it is also well-positioned to thrive under warmer ocean conditions. Conversely, *N. luetkeana* is likely to experience compounding negative effects of competition by increasing populations of *S. muticum*, decreased survival due to warming, and increased grazing by *P.gracilis* thus making them the most vulnerable of the three organisms we studied. Our study helps to identify relative vulnerabilities of interacting species within coastal food webs in the face of changing community structure and atmospheric heat waves.

## Funding

This research was supported by the Bates Hoffman Fellowship (Katrina H. Johnson) as well as Bates College Faculty Development Funds and the Friday Harbor Labs Patricia L. Dudley Fellowship (Katie A. Dobkowski). The funders had no role in study design, data collection and analysis, decision to publish, or preparation of the manuscript.

## Grant Disclosures

The following grant information was disclosed by the authors:

The Bates Hoffman Fellowship.

Bates College Faculty Development Funds.

Friday Harbor Labs Patricia L. Dudley Fellowship.

## Competing Interests

The authors declare there are no competing interests.

## Author Contributions

- Katrina H. Johnson conceived and designed the experiments, performed the experiments, analyzed the data, prepared figures and/or tables, authored or reviewed drafts of the article, and approved the final draft.
- Katie A. Dobkowski conceived and designed the experiments, performed the experiments, authored or reviewed drafts of the article, and approved the final draft.
- Sasha K. Seroy performed the experiments, authored or reviewed drafts of the article, and approved the final draft.
- Shelby Fox performed the experiments, authored or reviewed drafts of the article, and approved the final draft.
- Natalie Meenan performed the experiments, authored or reviewed drafts of the article, and approved the final draft.

## Data Availability

The raw data and code are available in the Supplementary Files.

## Supplemental Information

Supplemental information for this article can be found online at http://dx.doi.org/10.7717/peerj.15223#supplemental-information.

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
