# Peer review of "Feeding preferences and the effect of temperature on feeding rates of the graceful kelp crab, Pugettia gracilis"

_PeerJ, doi:10.7717/peerj.15223_

## Round 0.1 · original submission · Major Revisions

The design of this study is rigorous and the results are based on reliable and correct statistical methods. However, as the main point raised by the first reviewer, it is far-fetched to draw the present conclusions in this study with such a small sample size and study scale (within just one laboratory). I would therefore suggest that the authors refer to the suggestions of reviewers and make fundamental and substantial revisions to the discussion and conclusion sections to ensure that the extrapolation scale of the conclusions is not excessive.

Authors need to respond to and revise their manuscripts against the reviewers' comments one by one. I will seek additional reviewers' comments on the revised manuscript when I receive it.

·

Basic reporting

no comment

Experimental design

This study uses one small scale laboratory experiment to test weight change of an invasive and native seaweed in the presence of a crab herbivore, and another experiment to test weight change of the native seaweed in the presence of the crab at elevated and ambient temperatures. My main comment about the experimental design is that it is small scale and in several ways does not seem as comprehensive as necessary. Some specific points are:

1) On line 150-151 it states that “we used an unpaired t-test for the no-choice experiments and a paired t-test for the choice experiments”. I guess an unpaired t-test was used for the no-choice experiment because the trials were done at different times in different aquaria, while a paired t-test was done for the choice experiment because the both seaweed species were added to the same aquaria at the same time. However, even though the trials using each crab individual were done at different times in the no-choice experiment, the same individual crab consumers were still used in the different treatments, so the replicates are paired in that way, and I think a paired t-test should also have been used to analyse the data from the no-choice experiment.

2) About the temperature experiment, it is unclear why a control without the crabs was not used. Such a control was used in the food choice experiment but there is then no mention of any control being used in the experiment where feeding was done at different temperatures. So for the latter experiment, it is unknown if changes to seaweed weight were due to crab herbivory, or due to some other factor, such as natural weight loss in the aquarium conditions, etc.

3) Also, it is unclear in the temperature experiment why only one of the two seaweed species was used – it would have provided a more complete dataset to test temperature effects for both the seaweed species that were the overall focus of this study. It is possible for example that at ambient temperatures crabs prefer the native seaweed but at high temperatures that preference changes, but if any such interaction between food type and temperature exists it has not been explored using the limited experimental design of this study.

4) It does not seem that much attempt was made to simulate the natural benthic conditions in the laboratory aquaria, i.e. there is no mention of adding sand or rocks to simulate the natural benthos, and the crabs were tested separately so any inter or intraspecific effects, such as from competition or changed feeding behaviour in the presence of conspecifics, where not included in the experimental design.

Overall, this kind of laboratory experiment is the perfect situation to carefully test a range of treatments and their interactions in a unified way, for example, an experimental design could have been used that tested all together in the one experiment effects of the factors: type of algae (native alone, invasive alone, and both together), temperature (ambient and elevated), and presence of consumer (with a crab and without a crab). If this design was used, three-way anova could have tested the three factors and their interactions in the single analysis, instead of the three different t-tests done presently, and this would also reduce the potential for type-1 error. Using an experimental design like this would require capturing a much larger number of crabs, but this would be a good thing, as currently all the conclusions of the study were derived from the actions of only a small number of crabs (12 crabs in the food choice experiments and 18 in the temperature one) so the results do not have much potential to be generalised.

Validity of the findings

The authors consider that their findings from the temperature experiment can be used to make conclusions relevant to climate change (e.g. lines 28, 32, 92, 114, 260). They state (line 114, 235) that the time frame and range of their experimental temperature increases are relevant to episodic warming events that are increasing with climate change, but even so, I think it is a stretch to consider findings from a 48hr laboratory experiment done on 18 crabs to have much relevant connection to climate change. To make findings more relevant to climate change, even if wanting to know specifically about the increasing episodic warming events, measures could have been taken such as using long term experiments testing effects from present day vs predicted future frequency of recurrent episodic warming events, doing experiments in natural conditions in the field, or testing intergenerational adaptation to long-term temperature increases and to increased frequency of episodic warming events. Other comprehensive studies are already tackling these kinds of questions and producing findings that could be said to be relevant to climate change. For a small scale study like in the present manuscript, I would say that there could be a small amount of text about how the results may be considered in the context of climate change, but I do not think it should be emphasised so much e.g. in the abstract, as the final conclusion, etc.

Additional comments

Title: I think it would be useful in a small-scale study like this for it to be mentioned in the title that it was a laboratory study so the reader can see this straight away. Using the term "in the Salish Sea" makes it seem like the experiment was actually done in the sea, when in fact the only connection with the Salish Sea was that some small pieces of seaweed and 30 crabs were collected from that sea to use in the experiment in the laboratory.

line 58: it is good in papers about invasive species to add details also about where the invasive species were introduced from, i.e. in this case I'd be interested to know where is S. muticum's native range.

Reviewer 2 ·

Basic reporting

This paper investigates the feeding preferences of an important crab consumer in shallow subtidal ecosystems and the impacts of increased temperature on feeding rates on native kelp. This research is an important foundation for future work to explore the combined impacts of an invasive seaweed species and climate change on native kelp in a community ecology context. This paper includes a well-written introduction that is well-supported by appropriate literature review. Research questions and hypotheses are clearly stated, and their importance for climate change and invasion biology and community ecology are clearly explained. Likewise, the discussion is supported by the literature and important caveats and limitations of the study are addressed, with interesting suggestions for future research directions. This paper is self-contained, with results that are relevant to the presented hypotheses. While this study is an important contribution to the literature on the biological impacts of warming and invasive species, some discrepancies between the supplemental data files and results/methods in the text suggests that there could be some errors in the data analyses that need to be double-checked and potentially corrected (detailed below). Once corrections are made, it will be important to check that the results still support the conclusions/interpretations that have been made.

Experimental design

The research questions of this study are clearly defined and relevant to the fields of climate change, invasion, and community ecology. It is clear how this study contributes to the described knowledge gap.

The methods could be improved by adding more detail in the following areas:

Paragraph starting at line 161: Including the average (+/- SD) crab body sizes within each treatment would be helpful here. From the supplementary data, I calculated that the mean for ambient is ~3.2 and elevated is ~3.7 (please double check). Is this a statistically significant difference? (showing these results here would also be great) If so, the potential implications of larger crabs in the elevated treatment for conclusions should be discussed. Even though the authors are calculating food consumed per gram of crab, larger crabs could have physiological differences that those calculations do not account for.

Paragraph starting at line 161 continued: More details on the set-up for the temperature experiments would be helpful here. For example, was each crab held in its own tank with an independent heater and temperature controller? Or were multiple crabs held in a tank or tanks of each temperature treatment?

Validity of the findings

While raw data files have been provided, there are some discrepancies between the supplemental data files and results/methods in the text. This suggests that there could be some errors in the data analyses that need to be double-checked and potentially corrected. Once corrections are made, it will be important to check that the results still support the conclusions/interpretations that have been made. Suggestions for each file are provided below:

Supplementary 3 Table:
Shows 13 crabs for the “Ambient” treatment and 5 crabs for the “Elevated,” although the text states that there should be 12 and 6, respectively. I think crab 11 is the one that has the wrong treatment listed.

1KelpCrabData.csv file (Food preference experiments):
This file shows that for Crab 1, the NoChoice S. muticum trial is missing. For Crab 3, the Choice trial is missing. These missing data points should be explained in the methods. Since Crab 3 is missing from the Choice trials, the sample size is 11 instead of 12, thus the degrees of freedom should be 10 (the R script also shows a df of 10). However, the results still state that df = 11 (line 184). Please double-check these statistical analyses.

Line 181, No-choice experiment: After running the provided R script with the 1KelpCrabData.csv file, a Welch two-sample t-test resulted in a t-value of 0.75 and df of 20.028. The text here states t = 0.075 (a typo?) and df = 11. Please double-check the analyses and correct this discrepancy. Degrees of freedom for a Welch t-test are calculated differently! Also, the methods stated that an unpaired t-test was used for the no-choice experiment, but what was run in R was a Welch two-sample t-test. If the authors meant to do the unpaired two sample, they would have to specify ‘var.equal = TRUE’ (var.equal = FALSE, aka a Welch test, is the default in R). However, maybe the Welch test is more appropriate since the Crab 1 S. muticum trial is missing.

PaperMetric.csv file (Temperature experiments):
I believe the ‘Crab’ column in this file shows individual ID numbers? If so, these numbers do not correspond to the crab ID numbers in the Supplementary 3 table, which is a bit confusing.

The ‘Dates’ column indicates that these trials were conducted across two experiment periods - one starting in November and one in February? If this is the case, please describe this in the methods. For example, were crabs collected fresh from the field before each time frame, or were they all collected at the same time (before the November trials) and held in the lab? This would affect the length of the lab conditioning period for each time frame. Also please check the dates for typos!

In a couple trials, the kelp mass was higher at the end of the trial than at the beginning. Why might this be the case? Algal growth? How were these data points handled in analysis? Were they changed to zero, since it seems that the crab did not eat any of the kelp? Although the crab could have eaten some of the kelp, but growth overwhelmed that amount. Or were they included as is in the data file, negative values? I am not sure that negative values have any meaning here, since you can’t negatively eat. If these values might be due to error in measuring the kelp mass, they should be excluded.

R script:
It was not possible to replicate analyses for the temperature experiment. The code for making the plot includes “Mass” as a variable, which is not present in the PaperMetric.csv data file- although I think the equivalent variable in the file is “PaperMetric.” It is unclear if the data that are provided are the same data that were analyzed and presented in the results. Please also check assumptions of these tests, as was done for the choice/no choice experiments.

The R script also analyzes a “TEST.csv” file, which was not provided, so the t-tests cannot be run as presented.

I conducted a Welch’s t-test using the following code and the PaperMetric.csv file:

tempdataoutlier <- read.csv("PaperMetric.csv")
ambient <- filter(tempdataoutlier, Temp == "Ambient")
elevated <- filter(tempdataoutlier, Temp == "Elevated")
t.test(ambient$PaperMetric, elevated$PaperMetric)

This resulted in the following, which does not correspond to the results text:

Welch Two Sample t-test
data: ambient$PaperMetric and elevated$PaperMetric
t = -2.5643, df = 5.9533, p-value = 0.04296

Please double-check these analyses and make sure that the work can be replicated with the provided data and R script.

Additional comments

Small edits:

Line 43: Did the authors mean latitudinal instead of longitudinal?

Line 142: typos- “experiement” and “Crabs was”

---

## Round 0.2 · Minor Revisions

After I have carefully examined the reviewers' comments and your responses to them, I believe that the quality of the manuscripts has improved a lot. However, there is still some minor problems to be improved. In particular, there is a question about statistical methods in line 148. I don't think this is a major barrier, but it needs to be addressed in the new version.

·

Basic reporting

There are almost no spelling errors, typo's etc., please just on line 75 add the full stop, and on line 231 change "has" to "have".

Experimental design

line 148: if the assumption of homogeneity of variances has not yet been tested for the unpaired t-test used in the food choice experiment, please do this (e.g. using Levene's or Cochran's test) and present the outcome here or in the results. If this assumption has not been met, please try transforming the data or using a different test.

Validity of the findings

The validity of the food choice experiment may change if the data are needing to be transformed or analysed using a different test, based on the result of the test for homogeneity of variances described above.

---

## Round 0.3 · accepted · Accept

I checked the revised version submitted by the authors and checked the reviewers' suggestions. I believe that the current version has responded well to all the reviewers' concerns. The quality of the manuscripts has been substantially improved to meet the quality requirements of PeerJ.